# MULTI-VIEW DATA GENERATION WITHOUT VIEW SUPERVISION

**Mickaël Chen**
Sorbonne Université, CNRS, Laboratoire d'Informatique de Paris 6, LIP6, F-75005, Paris, France
`mickael.chen@lip6.fr`

**Ludovic Denoyer**
Sorbonne Université, CNRS, Laboratoire d'Informatique de Paris 6, LIP6, F-75005, Paris, France
Criteo Research
`ludovic.denoyer@lip6.fr`

**Thierry Artières**
Aix Marseille Univ, Université de Toulon, CNRS, LIS, Marseille, France
Ecole Centrale Marseille
`thierry.artiere@centrale-marseille.fr`

## ABSTRACT

The development of high-dimensional generative models has recently gained a great surge of interest with the introduction of variational auto-encoders and generative adversarial neural networks. Different variants have been proposed where the underlying latent space is structured, for example, based on attributes describing the data to generate. We focus on a particular problem where one aims at generating samples corresponding to a number of objects under various views. We assume that the distribution of the data is driven by two independent latent factors: the content, which represents the intrinsic features of an object, and the view, which stands for the settings of a particular observation of that object. Therefore, we propose a generative model and a conditional variant built on such a disentangled latent space. This approach allows us to generate realistic samples corresponding to various objects in a high variety of views. Unlike many multi-view approaches, our model doesn't need any supervision on the views but only on the content. Compared to other conditional generation approaches that are mostly based on binary or categorical attributes, we make no such assumption about the factors of variations. Our model can be used on problems with a huge, potentially infinite, number of categories. We experiment it on four image datasets on which we demonstrate the effectiveness of the model and its ability to generalize.

## 1 INTRODUCTION

Multi-view learning aims at developing models that are trained over datasets composed of multiple views over different objects. The problem of handling multi-view inputs has mainly been studied from the predictive point of view where one wants, for example, to learn a model able to predict/classify over multiple views of the same object (Su et al. (2015); Qi et al. (2016)). For example, using deep learning approaches, different strategies have been explored to aggregate multiple views but a common general idea is based on the (early or late) fusion of the different views at a particular level of a deep architecture. Few other studies have proposed to predict missing views from one or multiple remaining views as in Arsalan Soltani et al. (2017).

Recent research has focused on identifying factors of variations from multiview datasets. The underlying idea is to consider that a particular data sample may be thought as the mix of a content information (e.g. related to its class label like a given person in a face dataset) and of a side information, the view, which accounts for factors of variability (e.g. exposure, viewpoint, with/wo glasses...). All

samples of a given class share the same content information while they differ on the view information. A number of approaches have been proposed to disentangle the content from the view, also referred as the style in some papers (Mathieu et al. (2016); Denton & Birodkar (2017)). For instance, different models have been built to extract from a single photo of any object both the characteristics of the object but also the camera position. Once such a disentanglement is learned, one may build various applications like predicting how an object looks like under different viewpoints (Mathieu et al. (2016); Zhao et al. (2017)). In the generative domain, models with a disentangled latent space (Louizos et al. (2015); Edwards & Storkey (2015)) have been recently proposed with applications to image editing, where one wants to modify an input sample by preserving its content while changing its view (Lample et al. (2017); Kim et al. (2017b)) (see Section 6).

Yet most existing controlled generative approaches have two strong limitations: (i) they usually consider discrete views that are characterized by a domain or a set of discrete (binary/categorical) attributes (e.g. face with/wo glasses, the color of the hair, etc.) and could not easily scale to a large number of attributes or to continuous views. (ii) most models are trained using view supervision (e.g. the view attributes), which of course greatly helps learning such model, yet prevents their use on many datasets where this information is not available. Recently, some attempts have been made to learn such models without any supervision (Chen et al. (2016); Higgins et al. (2016)), but they cannot disentangle high level concepts as only simple features can be reliably captured without any guidance.

In this paper, we are interested in learning generative models that build and make use of a disentangled latent space where the content and the view are encoded separately. We propose to take an original approach by learning such models from multi-view datasets, where (i) samples are labeled based on their content, and without any view information, and (ii) where the generated views are not restricted to be one view in a subset of possible views. Following with our same example above, it means first, learning from a face dataset including multiple photos of multiple persons taken in various conditions related to exposure, viewpoint etc. and second, being able to generate an infinity of views of an imaginary person (or the same views of an infinity of imaginary persons) – see Figure 1. This contrast with most current approaches that use information about the style, and cannot generate multiple possible outputs.

More precisely, we propose two models to tackle this particularly difficult setting: a generative model (GMV - Generative Multi-view Model) that generates objects under various views (**multi-view generation**), and a conditional extension (C-GMV) of this model that generates a large number of views of any input object (**conditional multi-view generation**). These two models are based on the adversarial training schema of Generative Adversarial Networks (GAN) proposed in Goodfellow et al. (2014)). The simple but strong idea is to focus on distributions over pairs of examples (e.g. images representing *a same object* in *different views*) rather than distribution on single examples as we will explain later.

Our contributions are the following: (i) We propose a new generative model able to generate data with various *content* and high *view* diversity using a supervision on the content information only. (ii) We extend the model to a conditional model that allows generating new views over any input sample. (iii) We report experimental results on four different images datasets that show the ability of our models to generate realistic samples and to capture (and generate with) the diversity of views.

The paper is organized as follows. We first remind useful background on GANs and their conditional variant (Section 2). Then we successively detail our proposal for a generative multiview model (Section 3.2) and for its conditional extension (Section 4). Finally, we report experimental results on the various generative tasks allowed by our models in Section 5.2.

## 2 BACKGROUND

Our work is inspired by the *Generative Adversarial Network* (GAN) model proposed in Goodfellow et al. (2014). We briefly review the principle of GAN and of one of its conditional versions called *Conditional GAN* (CGAN) (Mirza & Osindero (2014)) that are the foundations of this work.

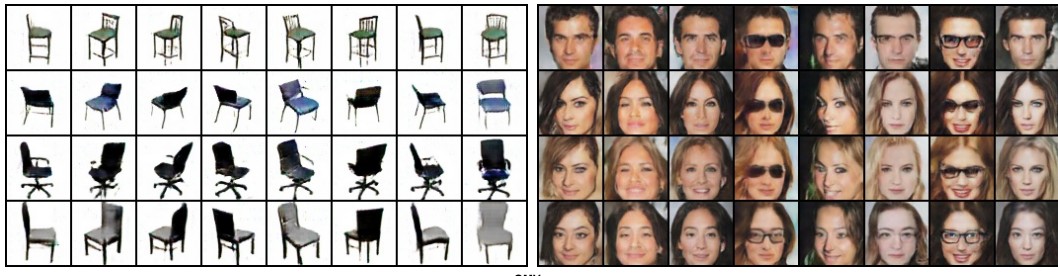

GMV

Figure 1: Samples generated by our model GMV on the *3DChairs* and on the *CelebA* datasets. All images in a row have been generated with the same content vector, and all images in a column have been generated with the same view vector.

## 2.1 GENERATIVE ADVERSARIAL NETWORK

Let us denote $\mathcal{X}$ an input space composed of multidimensional samples $\mathbf{x}$ e.g. vector, matrix or tensor. Given a latent space $\mathbb{R}^n$ and a prior distribution $p_{\mathbf{z}}(\mathbf{z})$ over this latent space, any generator function $G : \mathbb{R}^n \to \mathcal{X}$ defines a distribution $p_G$ on $\mathcal{X}$ which is the distribution of samples $G(\mathbf{z})$ where $\mathbf{z} \sim p_{\mathbf{z}}$. A GAN defines, in addition to G, a discriminator function $D : \mathcal{X} \to [0; 1]$ which aims at differentiating between *real* inputs sampled from the training set and *fake* inputs sampled following $p_G$, while the generator is learned to fool the discriminator D. Usually both G and D are implemented with neural networks. The objective function is based on the following adversarial criterion:

$$\min_G \max_D \mathbb{E}_{\mathbf{x} \sim p_{\mathbf{x}}} \left[ \log D(\mathbf{x}) \right] + \mathbb{E}_{\mathbf{z} \sim p_{\mathbf{z}}} \left[ \log(1 - D(G(\mathbf{z}))) \right] \tag{1}$$

where $p_{\mathbf{x}}$ is the empirical data distribution on $\mathcal{X}$.

It has been shown in Goodfellow et al. (2014) that if $G^*$ and $D^*$ are optimal for the above criterion, the Jensen-Shannon divergence between $p_{G^*}$ and the empirical distribution of the data $p_{\mathbf{x}}$ in the dataset is minimized, making GAN able to estimate complex continuous data distributions.

## 2.2 CONDITIONAL GENERATIVE ADVERSARIAL NETWORK

A conditional version of GAN (CGAN) has been proposed in Mirza & Osindero (2014). Instead of learning from an unsupervised dataset composed of datapoints $\mathbf{x}$, a CGAN is learned to implement a conditional distribution $p(\mathbf{x}|\mathbf{y})$ using a training set that consists of pairs of inputs/conditions $(\mathbf{x}, \mathbf{y})$ where $\mathbf{x}$ is a target and $\mathbf{y}$ is the condition. The conditionality of a CGAN is obtained by defining a generator function G that takes as inputs both a noise vector $\mathbf{z}$ and a condition $\mathbf{y}$. A target $\mathbf{x}$ from a given input $\mathbf{y}$ may be obtained by first sampling the latent vector $\mathbf{z} \sim p_{\mathbf{z}}$, then by computing $G(\mathbf{y}, \mathbf{z})$. The discriminator in a CGAN takes as inputs both the condition $\mathbf{y}$ and the (generated or real) datapoint $x$. It is learned to discriminate between *fake* input/target pairs (where the target is sampled using the generator) from *real* pairs drawn from the empirical distribution. The objective function can be written as:

$$\min_G \max_D \mathbb{E}_{(\mathbf{x}, \mathbf{y}) \sim p_{\mathbf{x}, \mathbf{y}}} \left[ \log D(\mathbf{x}, \mathbf{y}) \right] + \mathbb{E}_{\mathbf{z} \sim p_{\mathbf{z}}} \left[ \log(1 - D(G(\mathbf{y}, \mathbf{z}))) \right] \tag{2}$$

where $p_{\mathbf{x}, \mathbf{y}}$ stands for the empirical distribution on $(\mathbf{x}, \mathbf{y})$. As with regular GAN, with optimal G and D denoted respectively $G^*$ and $D^*$), the distribution $p_G^*$ fits the true empirical conditional distribution of the input/target pairs. Unfortunately, many studies have reported that on when dealing with high-dimensional input spaces, CGAN tends to collapse the modes of the data distribution, mostly ignoring the latent factor $\mathbf{z}$ and generating $\mathbf{x}$ only based on the condition $\mathbf{y}$, exhibiting an almost deterministic behavior. In most cases, CGAN is unable to produce targets with a satisfying diversity (see Section 5.2).

## 3 GENERATIVE MULTI-VIEW MODEL

### 3.1 OBJECTIVE AND NOTATIONS

We consider the multi-view setting where data samples represent a number of objects that have been observed in various views. The distribution of the data $\mathbf{x} \in \mathcal{X}$ is assumed to be driven by two latent factors: a **content** factor denoted $\mathbf{c}$ which corresponds to the invariant proprieties of the object, and a **view** factor denoted $\mathbf{v}$ which corresponds to the factor of variations. Typically, if $\mathcal{X}$ is the space of people's faces, $\mathbf{c}$ stands for the intrinsic features of a person's face while $\mathbf{v}$ stands for the transient features and the viewpoint of a particular photo of the face, including the photo exposure and additional elements like a hat, glasses, etc.... We assume that these two factors $\mathbf{c}$ and $\mathbf{v}$ are independent. This is a key property of the factors we want to learn.

We focus on two different tasks: (i) **Multi View Generation**: we want to be able to sample over $\mathcal{X}$ by controlling the two factors $\mathbf{c}$ and $\mathbf{v}$. Said otherwise, we want to be able to generate different views of the same object, or the same view of different objects. Given two priors, $\mathrm{p}(\mathbf{c})$ and $\mathrm{p}(\mathbf{v})$, this sampling will be possible if we are able to estimate $\mathrm{p}(\mathbf{x}|\mathbf{c}, \mathbf{v})$ from a training set. (ii) **Conditionnal Multi-View Generation**: the second objective is to be able to sample different views of a given object. For instance, it can be different views of a particular person based on one of his photos. Given a prior $\mathrm{p}(\mathbf{v})$, this sampling will be achieved by learning the probability $\mathrm{p}(\mathbf{c}|\mathbf{x})$, in addition to $\mathrm{p}(\mathbf{x}|\mathbf{c}, \mathbf{v})$.

The key issue for tackling these two tasks lies in the ability to accurately learn generative models able to generate from a disentangled latent space where the content factor and the view factor are encoded (and thus sampled) separately. This would allow controlling the sampling on the two different axis, the content and the view. The originality of our work is to learn such generative models without using any view labeling information.

Let us denote by $N$ the set of different objects and $n_i$ the number of views available for object number $i$ (not necessarily the same sets of views nor number of views for every object) such that $\{x_1^i, x_2^i, ..., x_{n_i}^i\}$ is the set of views for object $i \in [1; N]$. Moreover, we consider that $\mathbf{x}$ is a tensor (e.g. an image) and $\mathbf{c}$ and $\mathbf{v}$ are (latent) vectors in $\mathbb{R}^C$ and $\mathbb{R}^V$, $C$ and $V$ being the sizes of the content and view latent spaces. Note that this setting is not restrictive and corresponds, for instance, to categorization datasets where multiple photos of objects are available.

### 3.2 GENERATIVE MULTIVIEW MODEL

Let us consider two prior distributions over the content and view factors denoted as $\mathrm{p}_\mathbf{c}$ and $\mathrm{p}_\mathbf{v}$, these distributions typically being isotropic Gaussian distributions. These two distributions correspond to the prior distribution over content and latent factors. Moreover, we consider a generator $\mathrm{G}$ that implements a distribution over samples $\mathbf{x}$, denoted as $\mathrm{p}_\mathrm{G}$ by computing $\mathrm{G}(\mathbf{c}, \mathbf{v})$ with $\mathbf{c} \sim \mathrm{p}_\mathbf{c}$ and $\mathbf{v} \sim \mathrm{p}_\mathbf{v}$. Our objective is to learn this generator so that its first input $\mathbf{c}$ corresponds to the content of the generated sample while its second input $\mathbf{v}$, captures the underlying view of the sample. Doing so would allow one to control the output sample of the generator by tuning its content or its view (i.e. $\mathbf{c}$ and $\mathbf{v}$).

Yet it is expected that learning $\mathrm{G}$ using a standard GAN approach would not allow to accurately disentangle the latent space. Indeed, without constraint, the content and view factors are going to be diluted in the input latent factor $\mathbf{z}$ of the GAN, with no possibility to know which dimensions of $\mathbf{z}$ capture the content and which capture the view factor. We propose a novel way to achieve this desired feature.

The key idea of our model is to focus on the distribution of *pairs* of inputs rather than on the distribution over individual samples. We explain now which pairs we are talking about and why considering pairs might be useful.

First, **which pairs are we considering?** When no view supervision is available the only valuable pairs of samples that one may build from the dataset consist of two samples of a given object under two different views. Indeed, choosing randomly two samples of a given object will most of the time correspond to different views of this object, especially when considering continuous views as we do. Fortunately, considering a generator $\mathrm{G}$ as discussed above (operating on a couple of a content

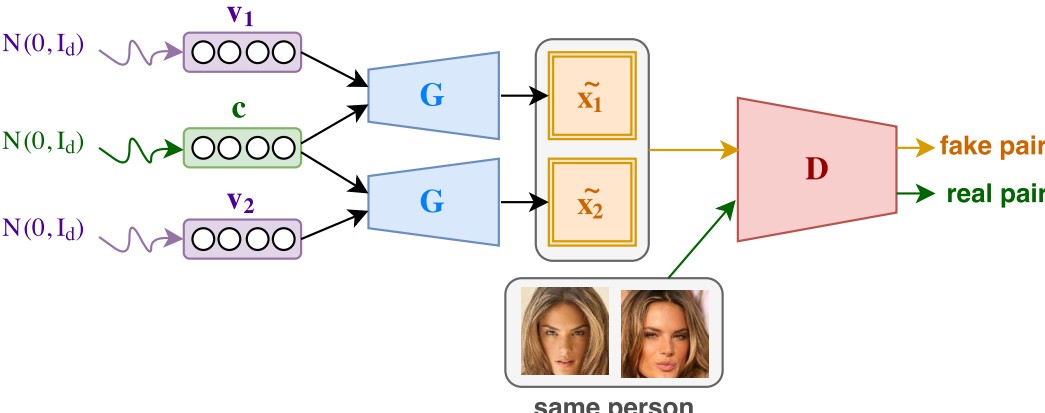

Figure 2: Overview of the GMV model. The generator G produces an image given a content vector **c** and view vector **v**. A pair of images is generated by sampling a common content factor $\mathbf{c} \sim p_{\mathbf{c}}$ but two different views factors $\mathbf{v_1} \sim p_{\mathbf{v}}$ and $\mathbf{v_2} \sim p_{\mathbf{v}}$. The discriminator D is learned to distinguish between such pairs of generated images and real pairs of samples corresponding to a same object under different views. Real pairs are built by choosing at random two training samples of the same object. Those samples should most of the time correspond to two different views. No information on the views is used here.

vector **c** and of a view vector **v**), one can generate corresponding artificial (fake) pairs of samples by sampling a single content vector $\mathbf{c} \sim p_{\mathbf{c}}$ to combine with two different view vectors $\mathbf{v_1} \sim p_{\mathbf{v}}$ and $\mathbf{v_2} \sim p_{\mathbf{v}}$, i.e. constructing $G(\mathbf{c}, \mathbf{v_1})$ and $G(\mathbf{c}, \mathbf{v_2})$ – see Figure 2.

Second, **why working on such pairs would be interesting?** Following the GAN idea, we propose to learn a discriminator to distinguish between such real and fake pairs. To be able to fool the discriminator, the generator then has to achieve three goals. (i) As in regular GAN, each sample generated by G needs to look realistic. (ii) Moreover, because real pairs are composed of two views of the same object, the generator should generate pairs of the same object. Since the two sampled view factors $\mathbf{v_1}$ and $\mathbf{v_2}$ are different, the only way this can be achieved is by encoding the invariant features into the content vector **c**. (iii) Finally, it is expected that the discriminator should easily discriminate between a pair of samples corresponding to the same object under different views from a pair of samples corresponding to a same object under the same view. Because the pair shares the same content factor **c**, this should force the generator to use the view factors $\mathbf{v_1}$ and $\mathbf{v_2}$ to produce diversity in the generated pair.

The Generative Multiview Model's (GMV) architecture is detailed in Figure 2. It is learned by optimizing the following adversarial loss function:

$$\min_{G} \max_{D} \mathbb{E}_{\mathbf{x_1},\mathbf{x_2} \sim p_{\mathbf{x}}|l(\mathbf{x_1})=l(\mathbf{x_2})}[\log D(\mathbf{x_1}, \mathbf{x_2})] + \mathbb{E}_{\mathbf{v_1},\mathbf{v_2} \sim p_{\mathbf{v}},\mathbf{c} \sim p_{\mathbf{c}}}[\log(\mathbf{1} - \mathbf{D}(G(\mathbf{c}, \mathbf{v_1}), G(\mathbf{c}, \mathbf{v_2})))] \tag{3}$$

where $l(\mathbf{x})$ stands for the label of **x** (e.g. a particular person in a face dataset). Note that, since the proposed model can be seen as a particular GAN architecture over pairs of inputs, the global minimum of the learning criterion is obtained when the model is able to sample pairs of views over a similar object.

**Using the Model at inference:** As discussed above the training of the discriminator on pairs of samples introduce useful constraints on how the content and the view information are used to generate samples. Once the model is learned, we are left with a generator $G$ that generates single samples by first sampling **c** and **v** following $p_{\mathbf{c}}$ and $p_{\mathbf{v}}$, then by computing $G(\mathbf{c}, \mathbf{v})$. By freezing **c** or **v**, one may then generate samples corresponding to multiple views of any particular content, or corresponding to many contents under a particular view. One can also make interpolations between two given views over a particular content, or between two contents using a particular view (See examples in Figure 4).

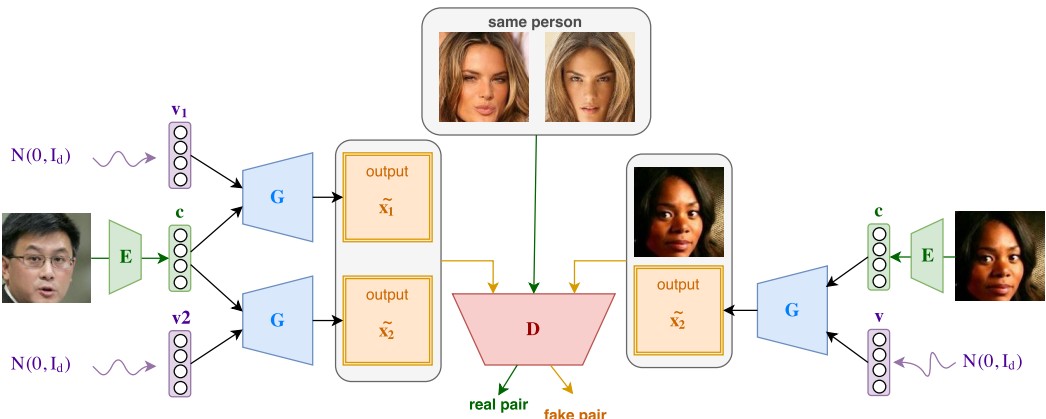

Figure 3: The conditional generative model C-GMV. Content vectors are not randomly sampled anymore, but are induced from real inputs through an encoder E. The discriminator is provided two types of negative examples: examples of the first type are pairs of generated samples using a same content factor but with two different views (left). The second type of negative examples is composed of pairs of a real sample $\mathbf{x}$ and of a generated sample built from $\mathbf{x}$ using a CGAN like approach (right). This artificial sample corresponds to the same content as in input sample $\mathbf{x}$ but under a different view. Note that the left part of the architecture is crucial for the model to take into account the view factor, and thus to generate diversity which cannot be obtained using the C-GAN component only.

## 4 CONDITIONAL GENERATIVE MODEL (C-GMV)

The GMV model allows one to sample objects with different views, but it is not able to change the view of a given object that would be provided as an input to the model. This, however, might be of interest in particular applications like image editing. This section aims at extending our generative model the ability to extract the content factor from any given input and to use this extracted content in order to generate new views of the corresponding object. To achieve such a goal, we must add to our generative model an encoder function denoted $E : \mathcal{X} \rightarrow \mathbb{R}^C$ that will map any input in $\mathcal{X}$ to the content space $\mathbb{R}^C$ (see Figure 3).

To do so we take inspiration from the CGAN model (Section 2.2). We will encode an input sample $\mathbf{x}$ in the content space using an encoder function, noted $E$ (implemented again as a neural network). This encoder serves to generate a content vector $\mathbf{c} = E(\mathrm{x})$ that will be combined with a randomly sampled view $\mathbf{v} \sim \mathrm{p}_{\mathbf{v}}$ to generate an artificial example. The artificial sample is then combined with the original input $\mathbf{x}$ to form a negative pair. This is illustrated in the extreme right part of Figure 3 which exactly corresponds to a CGAN architecture. Yet CGAN has severe weaknesses and are known to easily miss modes of the underlying distribution. The generator enters in a state where it ignores the noisy component $\mathbf{v}$ (see results in Figure 7). To overcome this phenomenon, we use the same idea as in GMV. We build negative pairs $(\mathrm{G}(\mathbf{c}, \mathbf{v_1}), \mathrm{G}(\mathbf{c}, \mathbf{v_2}))$ by randomly sampling two views $\mathbf{v_1}$ and $\mathbf{v_2}$ that we combine to a unique content $\mathbf{c}$. This time, however, $\mathbf{c}$ is not sampled according to a noise distribution but it is computed from a sample $\mathbf{x}$ using the encoder $\mathbf{E}$, i.e. $\mathbf{c} = E(\mathbf{x})$.

By doing so, we preserve the ability of our approach to generating pairs with view diversity. Since this diversity can only be captured by taking into account the two different view vectors provided to the model ($\mathbf{v_1}$ and $\mathbf{v_2}$), this will encourage $\mathrm{G}(\mathbf{c}, \mathbf{v})$ to generate samples containing both the content information $\mathbf{c}$, and the view $\mathbf{v}$. As it was done for the GMV model, positive pairs are sampled from the training set and correspond to two views of a given object.

| Dataset | number of samples | | Number of objects | | Views per object | | |
|---|---|---|---|---|---|---|---|
| | train | test | train | test | min | mean | max |
| CelebA (Liu et al. (2015)) | 198791 | 3808 | 9999 | 178 | 1 | 19.9 | 35 |
| 3DChairs (Aubry et al. (2014)) | 80600 | 5766 | 1300 | 93 | 62 | 62 | 62 |
| MVC cloth (Liu et al. (2016)) | 159128 | 2132 | 37004 | 495 | 4 | 4.3 | 7 |
| 102flowers (Nilsback & Zisserman (2008)) | 8189 | – | 102 | – | 40 | 80.3 | 258 |

Table 1: Datasets Statistics: Train and Test data include samples corresponding to different objects. GMV and CGMV are trained on the *train* part. The test part contains the images that are used as inputs of CGMV and CGAN.

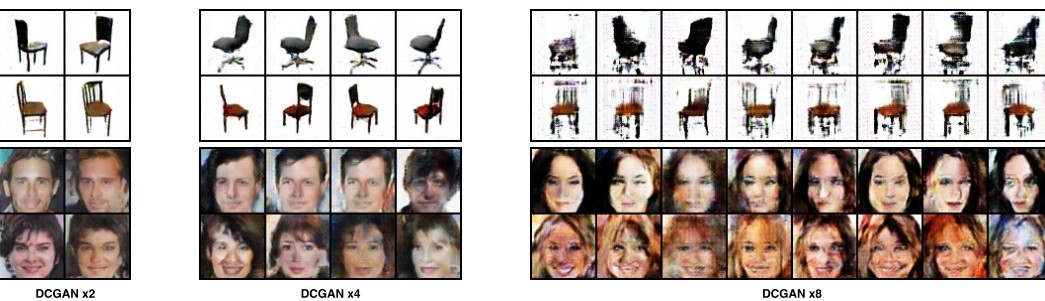

Figure 4: Samples generated by the DCGANx2, DCGANx4 and DCGANx8 models. These samples have to be compared to the ones presented in Figure 1.

In this setting, the resulting adversarial loss function can be written as:

$$\min_{G} \max_{D} \quad \mathbb{E}_{\mathbf{x_1},\mathbf{x_2} \sim p_{\mathbf{x}} | l(\mathbf{x_1}) = l(\mathbf{x_2})} [\log D(\mathbf{x_1}, \mathbf{x_2})]$$
$$+ \mathbb{E}_{\mathbf{v_1},\mathbf{v_2} \sim p_{\mathbf{v}}, \mathbf{x} \sim p_{\mathbf{x}}} [\log(1 - D(G(E(\mathbf{x}), \mathbf{v_1}), G(E(\mathbf{x}), \mathbf{v_2})))]$$
$$+ \mathbb{E}_{\mathbf{v} \sim p_{\mathbf{v}}, \mathbf{x} \sim p_{\mathbf{x}}} [\log(1 - D(G(E(\mathbf{x}), \mathbf{v}), \mathbf{x}))]$$

At inference time, as we did with the GMV model, we are interested in getting the encoder E and the generator G. These models may be used for generating new views of any object which is observed as an input sample $\mathbf{x}$ by computing its content vector $E(\mathbf{x})$, then sampling $\mathbf{v} \sim p_{\mathbf{v}}$ and finally by computing the output $G(E(\mathbf{x}), \mathbf{v})$.

## 5 EXPERIMENTAL RESULTS

### 5.1 EXPERIMENTAL PROTOCOL

**Datasets:** In order to evaluate the quality of our two models, we have performed experiments over four image datasets of various domains. The statistics of the dataset are given in Table 1. Note that when supervision is available on the views (like CelebA for example where images are labeled with attributes) we do not use it for learning our models. The only supervision that we use is if two samples correspond or not to the same object.

**Model Architecture:** We have used the same architectures for every dataset. The images were rescaled to $3 \times 64 \times 64$ tensors. The generator G and the discriminator D follow that of the DCGAN implementation proposed in Radford et al. (2015). For the conditional model, the encoder E is similar to D except for the first layer that includes a batch normalization and the last layer that doesn't have a non-linearity. Following the article, an implementation of our algorithms is freely available[1].

Learning has been made using classical GAN learning techniques: we used Adam optimizer (Kingma & Ba (2014)) with batches of size 128. Following standard practice, learning rate in the

---

[1] https://github.com/mickaelChen/GMV

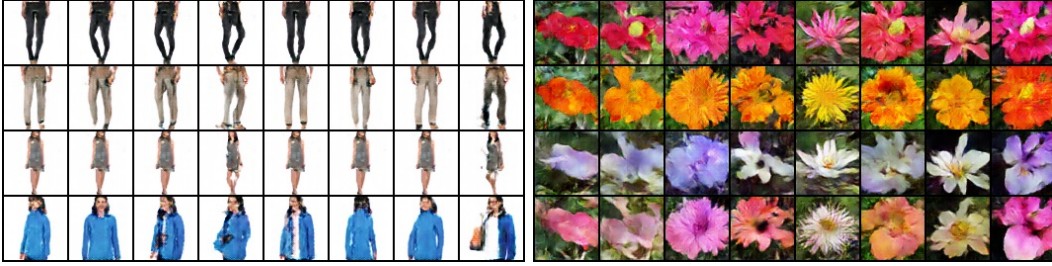

Figure 5: Samples generated by the GMV model on the *MVC Cloth* and on the *102flowers* datasets by GMV model. All images in a row have been generated with the same content vector, and all images in a column have been generated with the same view vector.

GMV experiments are set to $1 \cdot 10^{-3}$ of G and $2 \cdot 10^{-4}$ for D. For the C-GMV experiments, learning rates are set to $5 \cdot 10^{-5}$. The adversarial objectives are optimized by alternating gradient descent over the generator/encoder, and over the discriminator.

**Baselines:**    We compare our proposal with recent state-of-the-art techniques. However, most existing methods are learned on datasets with view labeling. To fairly compare with alternative models we have built baselines working in the same conditions as our models. In addition we compare our models with the model from Mathieu et al. (2016). We report results gained with two implementations, the first one based on the implementation provided by the authors[2] (denoted Mathieu et al. (2016)), and the second one (denoted Mathieu et al. (2016) (DCGAN) ) that implements the same model using architectures inspired from DCGAN Radford et al. (2015), which is more stable and that we have carefully tuned to allow a fair comparison with our approach.

For pure multi-view generative setting, we compared our generative model (GMV) with standard GANs that are learned to approximate the joint generation of multiple samples: DCGANx2 is learned to output pairs of views over the same object, DCGANx4 is trained on quadruplets, and DCGANx8 on eight different views. To be more detailed, for instance the generator of a DCGANx2 model takes as input a sampled noise vector and outputs 2 images, while its discriminator is learned to distinguish these pairs of images as negative samples from positive samples which are built as for the GMV model, i.e. pairs of samples in the dataset that corresponds to the same object. The main difference with GMV is that the above GAN-based methods do not explicitly distinguish content and view factors as it is done in GMV.

Likewise, for conditional generation, we compared our approach *C-GMV* with CGAN models that we briefly introduced in Section 2.2 and with the two variants of the model from Mathieu et al. (2016) that we just mentioned.

## 5.2    EXPERIMENTAL RESULTS

### GENERATING MULTIPLE CONTENTS AND VIEWS

We first evaluate the ability of our model to generate a large variety of object and views. Figure 1 shows examples of generated images by our model and Figure 4 shows images sampled by DCGAN-based models (DCGANx2, DCGANx4, and DCGANx8) on *3DChairs* and *CelebA* datasets (more results are provided in the Appendix). For GMV generated images, a row shows a number of samples that have been generated with the same sampled content factor $\mathbf{c} \sim p_{\mathbf{c}}$ but with various sampled view factors $\mathbf{v} \sim p_{\mathbf{v}}$, while the same view factor $\mathbf{v}$ is used for all samples in a column. Figure 5 shows additional results, using the same presentation, for the GMV model only on two other datasets.

One sees on these figures that our approach allows to accurately generate the same views of multiple objects, or alternatively the multiple views of a single object. The generated images are of good

---

[2]https://github.com/MichaelMathieu/factors-variation

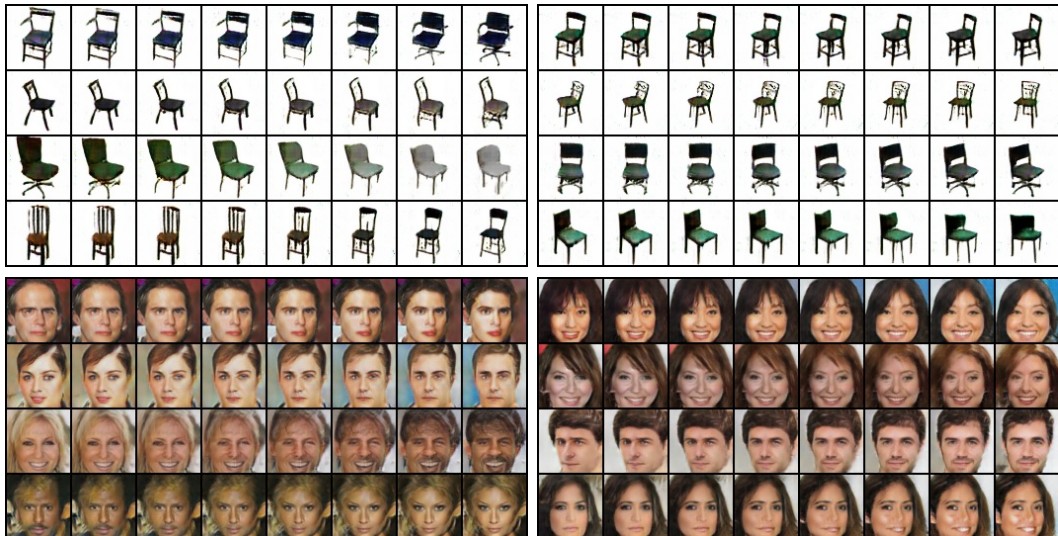

Figure 6: Samples generated by the GMV model by using interpolation on the content (left) or on the view (right) for *3DChairs* (up) and *CelebA* datasets (bottom). Within each of the four boxes, each row is independent of the others. For the two left boxes: The left and right column correspond to generated samples with two sampled content factors, while the middle images correspond to the samples generated by using linear interpolated content factors between the two extreme content factors while the view factor remains fixed. The two right boxes are built the same way by exchanging the roles of view and content.

quality, and the diversity of the generated views is high, showing the ability of our model to capture the diversity of the training dataset in terms of possible views.

Figure 4 shows similar results for GAN-based models. For images generated by these models a row corresponds to the multiple images produced by the model for a given sampled noise vector. When comparing GMV generated images to those generated by GAN-based models, one can see that the quality of images produced by DCGANx2 is comparable to the ones we obtain with GMV showing our approach has the same ability as a GAN to generate good outputs. But DCGANx2 is only able to generate two views of each object since it does not distinguish content and view factors. For the same reason, the images in the same column (for different objects) do not necessarily match the same view. While DCGANx4 or DCGANx8 could have the ability to generate more views for each object, the learning problem is more difficult due to the very high-dimensionality of the observation space, and the visual qualities of the generation degrade.

Figure 6 shows generated samples obtained by interpolation between two different view factors (left) or two content factors (right). It allows us to have a better idea of the underlying view/content structure captured by GMV. We can see that our approach is able to smoothly move from one content/view to another content/view while keeping the other factor constant. This also illustrates that content and view factors are well independently handled by the generator i.e. changing the view does not modify the content and vice versa.

GENERATING MULTIPLE VIEWS OF A GIVEN OBJECT

The second set of experiments evaluates the ability of C-GMV to capture a particular content from an input sample, and to use this content to generate multiple views of the same object. Figure 7 and 8 illustrate the diversity of views in samples generated by our model and compare our results with those obtained with the CGAN model and to models from Mathieu et al. (2016). For each row the input sample is shown in the left column. New views are generated from that input and shown on the right. Concerning the CGAN approach, the mode collapse phenomenon that we previously described clearly occurs: the model does not take into account the view factor and always generate similar samples without any view diversity. The C-GMV model demonstrates here that it is able to

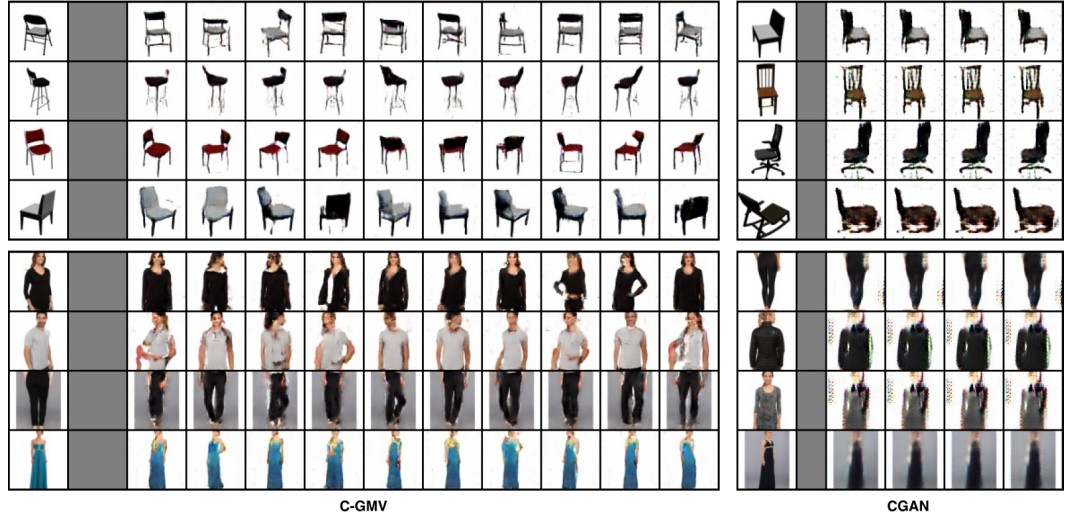

Figure 7: Samples generated by conditional models. The images on the left are generated by C-GMV while the images on the right are generated by a single CGAN. The leftmost column corresponds to the input example from which the content factor is extracted, while other columns are images generated with randomly sampled views.

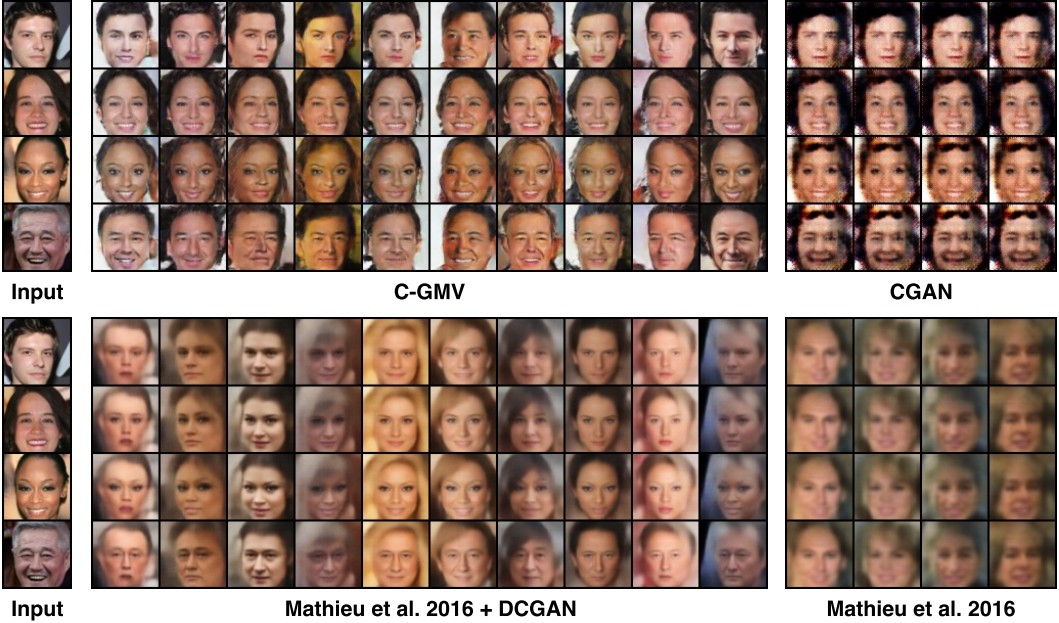

Figure 8: Samples generated by conditional models: CGMV and CGAN (top), Mathieu et al. (2016) and DCGAN variant (bottom). The figure shows samples generated from four input images (leftmost column) by computing their content factors and by randomly sampling one view factor per column.

| Dataset | Number of objects | Acc on test | Acc on $E(x)$ | Acc on $G(E(x), v)$ with $v \sim p_v$ |
|---------|-------------------|-------------|---------------|----------------------------------------|
| 3D chairs | 93 | 96.7% | 93.6% | 61.3% |
| MVC cloth | 495 | 45.2 % | 31.5% | 27.1% |

Table 2: Classification results: *Acc* is the accuracy of the classifier on test images, encoded images, and generated views.

extract the content information, and to generate a large variety of views of any object for which we have one image.

One can see that the quality of the CGMV images clearly outperforms the quality of CGAN images. Moreover, mode collapse of CGAN can be observed, making this model unable to generate diversity. Finally, comparison with the closest work from Mathieu et al. (2016) shows that images generated by the CGMV model have a better quality and are more diverse.

To do so, we estimate the performance of simple classifiers operating on images, either true samples or generated. We consider two subsets of objects $S_1$ and $S_2$ with a null intersection. We then train a C-GMV model on training samples from $S_1$. We then use this model on test samples from $S_2$, yielding both generated images corresponding to objects in $S_2$ but with new views, and content vectors for these images. We then evaluate the performance of two classifiers. One is learned and tested on content vectors. The other one is trained on real images and tested on generated images. The latter is also evaluated on real images on the test set of $S_2$ for reference.

Table 2 reports such classification results obtained over two of the studied datasets. On these two datasets, one can see that the accuracy of the classifiers operating on true images and on content latent factors are close, showing that our model has actually captured the category information in $E(x)$ as it is desired. Moreover, although the accuracy of the classifier learned on real images is lower when computed on generated samples, which is, of course, expected, the drop of performances seems reasonable and shows that C-GMV is able to reconstruct the content information well.

## 5.3 EVALUATION OF THE QUALITY OF THE GENERATED SAMPLES

The next sets of experiments aimed at evaluating the quality of the generated samples. They have been made on the CelebA dataset and evaluate (i) the ability of the models to preserve the identity of a person in multiple generated views, (ii) to generate realistic samples, (iii) to preserve the diversity in the generated views and (iv) to capture the view distributions of the original dataset.

### IDENTITY PRESERVATION

We propose a set of experiments to measure the ability of our model to generate multiple views of the same image i.e. **to preserve the identity of the person** in two generated images. In order to evaluate if two images belong to the same person, we extracted features using VGG Face descriptor (Parkhi et al.), which has been proposed for person reidentification task. The proximity between the face descriptors computed from two images reflects the likeness of the persons in these images. Tables 3 and 4 illustrate the AUC obtained by the different models. This AUC corresponds to the probability that a positive pair (i.e. a pair with two images of the same person) is associated with a lower distance than a negative pair. This AUC has been computed using 15 000 positive pairs and 15 000 negative ones. An AUC of $100\%$ corresponds to a perfect model. We compute the AUC using generated positive pairs and generated negative pairs as a global estimation of the quality of our model. We also compute two additional AUC score by comparing generated positive pairs with negative pairs sampled from the dataset, and comparing real positive pairs with generated negative pairs, to better characterize the behavior of the models.

First, one can see that when pairs of images are sampled directly from the dataset (Table 3, first column), our classifier obtains an AUC of 93.3% (and not 100 %). This is due to the imperfection of the identity matching model that we use. This value can thus be seen as an upper bound of the expected performance of the different models. When comparing generative models (Table 3, columns 2 to 5), the GMV model obtains an AUC of 76.3 % which significantly outperforms the other generative models (GANx2, GANx4, and GANx8) that are less able to generate images of the

| positive pair | | negative pair | CelebA | GMV | GANx2 | GANx4 | GANx8 |
|---|---|---|---|---|---|---|---|
| generated | vs | generated | 93.3 % | 76.3 % | 72.0 % | 74.2 % | 57.5% |
| generated | vs | real | 93.3 % | 92.9 % | 98.2 % | 99.1 % | 99.9% |
| real | vs | generated | 93.3 % | 78.7 % | 51.5 % | 47.8 % | 13.3% |

Table 3: Identity preservation AUC of **generative models**. The *CelebA* column is the AUC obtained when positive pairs correspond to two images of the same person sampled from the test set, and negative pairs to images of two different persons.

| positive pair | | negative pair | CGMV | Mathieu (2016) | Mathieu (2016) (DCGAN) | CGAN |
|---|---|---|---|---|---|---|
| input and generated | vs | generated pair | 67.2% | 50.6% | 77.3 % | 69.8 % |
| generated pair | vs | generated pair | 75.1% | 61.2 % | 84.2 % | 100 % |

Table 4: Identity preservation AUC of **conditional models**. The first line corresponds to the distance computed between a real image, and an image generated using this real image as an input (positive pairs) or another image (negative pairs). The second line compares two generated images based on the same input image (positive pairs) or based on two different images (negative pairs).

same person. It means that 76.3 % of the pairs generated with the same content vector $c$ (but random view vectors) have a lower distance than pairs generated with two randomly sampled content vectors $c_1$ and $c_2$. More specifically, while all models are able to consistently generate positive pairs (Table 3 line 2), only the GMV can reliably generate negative pairs (Table 3 line 3). This hints at a lack of diversity in the image generated by the other models.

When comparing conditional models, the pair of images is generated based on a given input image. The AUC thus measures the ability of the models to generate new views while preserving the identity of the input person. We compare the CGMV model with the two implementations of the model by Mathieu et al. (2016) that we mentioned previously.

One can see on Table 4 (first line) that the Mathieu et al. (2016) (DCGAN) model outperforms our proposed CGMV model, obtaining an AUC of 77.3 % while our approach obtains 67.2 %. It thus means that Mathieu et al. (2016) is better to generate a view of an input image while preserving the identity of the person. When comparing two generated images, the AUC of Mathieu et al. (2016) is 85.4% while CGMV obtains an AUC of 76.1%. At first glance, it gives the impression that Mathieu et al. (2016) is better than our approach. Yet, looking at the CGAN score, one can see that this last model obtains a 100% AUC (second line). In fact, this score is due to the fact that the CGAN model is unable to generate diversity in the generated views, and thus generates two images that are exactly the same and that thus are easy to classify as belonging to the same person. This means that we also have to evaluate the different models in terms of **output quality** (evaluate if the generated images are of good quality), and in terms of **diversity** (i.e. the generated views are different, and accurately capture the dataset distribution). This is the focus of the next sections.

QUALITY AND DIVERSITY OF THE GENERATED IMAGES

To deeply evaluate the quality of generative models we studied the distribution of generated samples by our models and by the baselines on the Celeba dataset. The idea is that a good generative model should generate samples with a distribution on attributes that is close to the one in the training set (while it is not actually included in the objective criterion).

A first measure of the quality of the generated samples is based on the **blurry** attribute of the CelebA dataset. This attribute identifies blurry images in the dataset i.e. images that are less realistic. We have used this attribute to train a blurry-detection classifier that has been used to evaluate the quantity of blurry generated images (Table 5), the idea being that a good method would generate the same proportion of blurry samples than the proportion of blurry images in the dataset. The results are illustrated in Table 5. While the Blurry attribute has probability 0.05 according to the true empirical distribution (this probability being estimated as 0.08 using attribute classifiers), it goes up to more than 0.99 with CGAN and Mathieu et al. (2016) and it is above 0.5 for all GANx

| CGAN | GANx2 | GANx4 | GANx8 | Mathieu (2016) | Mathieu (2016) (DC-GAN) | GMV | CGMV | celebA Dist. | Real Dist. |
|---|---|---|---|---|---|---|---|---|---|
| 99.8 % | 52.5 % | 82.4 % | 98.6 % | 99.9 % | 74% | 28.9% | 37.2 % | 8.61% | 5.1% |

Table 5: Evaluation of the blurry character of generated images. The "CelebA' column is the value obtained by the Blurry-detection classifier on the test set, while the "Real Dist.' column corresponds to the proportion of images labeled as "blurry" in the original test dataset.

models and about 0.75 for Mathieu et al. (2016) (DCGAN), while it remains limited at most at 0.4 for GMV and CGMV. GMV and CGMV models clearly generate less blurry images than the GAN/CGAN/Mathieu et al. (2016) models giving the intuition that the generated images are more realistic using our technique. Even if the evaluation method is imperfect, it tends to assess that the CGMV and GMV methods are better able to generate interesting outputs.

We also studied the distribution of generated samples by our models and by our baselines on 40 binary attributes of the CelebA dataset. To understand the behavior of a generative model we generated a set of samples with it, from which we computed the distribution of generated samples with respect to each of the 40 attributes (see Table 8), and we finally computed a distance between this distribution and the true dataset distribution. Note that, as the ground truth on attributes is not available for generated samples, we used attribute labels provided by classifiers, one for every attribute, that have been learned on the Celeba training set to infer the values of the 40 attributes of a sample. The distribution of generated samples is then estimated by using these classifiers and data generated from 3800 samples corresponding either to the 3800 test samples with randomly sampled view vector (for conditional models such as CGMV and Mathieu) or to 3800 randomly generated samples for pure generative models (such as GMV). The estimated true distribution is estimated by using the 40 classifiers on 3800 test samples, it is noted the *Celeba* distribution hereafter. For information we also provide the true empirical distribution (abusively noted as *Real distribution* hereafter) computed from the available attribute labels of the test samples. All estimated distributions are detailed in the supplementary material.

We observed that many rare attributes were completely ignored (i.e. no occurrence in generated samples) by models such as Mathieu et al. (2016), CGAN, GANx8 and that, on the contrary, common attributes were sometimes over generated by such models (e.g. Young). Globally one sees strong differences between models with GMV and CGMV seeming to better capture the dataset distribution, which next statistics reveal better.

Next, we computed a score for each generative model that is based on these distributions. For each model and for each attribute we computed the Bhattacharyya distance (with value in $[0, \infty]$) between the distribution yielded by a model and the estimated true distribution. We report the sum over attributes of these distances in Table 6. As may be seen, the distance computed between the estimated true distribution and the ones yielded by models GMV and CGMV are very low, below 0.1, actually lower than the distance between the true empirical distribution and the estimated true distribution. Moreover the distance computed for Mathieu et al. (2016) and Mathieu et al. (2016) (DCGAN) models, CGAN and all GANx models are clearly higher than those of our models.

To conclude on this set of experiments, the GMV and CGMV models seem to be the best trade-off between identity preservation and diversity in the generated views. While Mathieu et al. and CGAN tend to have a better identity preservation ability, it is at the price of generating samples with a much lower diversity than our approach which is the most able to capture the dataset distribution.

The above results show clear differences between GMV and CGMV models and state of the art methods such as Mathieu et al. (2016) and CGAN. Yet it does not fully inform about the diversity of generated samples which is a key feature for such generative models, either purely generative or conditional. We conducted the following experiment which aims at evaluating the diversity through the number of unique combinations of the 40 attributes that occur in a set of generated samples. Again to fairly compare results we use our attribute classifiers learned on the training set to label sample images according to the 40 attributes. Looking at the true distribution of the test samples there are

|        | CGAN | GANx2 | GANx4 | Mathieu (2016) | Mathieu (2016) (DC-GAN) | GMV | CGMV | celebA Dist. | Real Dist. |
|--------|------|-------|-------|----------------|-------------------------|-----|------|--------------|------------|
| D2T    | 1.8  | 0.66  | 0.74  | 1.97           | 0.89                    | 0.31 | 0.42 | 0            | 0.25       |
| D2E    | 1.3  | 0.25  | 0.43  | 1.37           | 0.39                    | 0.06 | 0.09 | 0.25         | 0          |

Table 6: Bhattacharyya distance between attributes distribution of generated samples by few generative models and the true empirical distribution (Real Distribution) in line *D2T* or the Estimated distribution (Celeba) in line *D2E* (see text) computed by the use of the 40 classifiers over the dataset. The higher the distance is, the lesser the model is able to capture the distribution of the different attributes.

| Mathieu et al. (2016) | Mathieu et al. (2016) (DCGAN) | CGMV | GMV | CGAN | CelebA Dist. | Real Dist. |
|-----------------------|-------------------------------|------|-----|------|--------------|------------|
| 3%                    | 19%                           | 43%  | 46% | 9%   | 47%          | 57 %       |

Table 7: Ratio of unique combinations of attributes in generated images (see text). GMV and CGMV are the only models able to generate a diversity close to the diversity observed in the dataset.

about 1795 unique combinations amongst the 3800 test samples, i.e. a ratio of 47 % of unique combinations (which is actually 57 % on the 200k samples of the training set). For conditional models we generated samples by computing the content vector of the 3800 test samples, then by randomly sampling 3800 view vectors and finally generating 3800 samples. We then computed the attribute labels of these samples and count the number of unique combinations of the 40 attributes amongst the 3800 images. For pure generative models we randomly sampled 3800 pairs of content and view vectors and computed the same statistic (note that results for pure generative and conditional models are not fully comparable since a number of test samples correspond to a same individual). Table 7 compares the obtained results. These results clearly show the behavior of CGAN and Mathieu et al. (2016) models which fully ignore a number of rare attributes (see also Table 8 in Appendix) hence yield very poor diversity. While Mathieu et al. (2016) (DCGAN) is better than the original Mathieu et al. (2016) model, it does not reach the diversity obtained with either GMV and CGMV models which are close to the estimated true empirical diversity.

## 6 RELATED WORK

Learning generative models using three sets of latent variables to describe a pair of objects has been proposed a long time ago and is known as inter-battery factor analysis (IBFA) Tucker (1958); Klami et al. (2013) . Such methods are very related to Canonical Correlation Analysis (CCA) and have been used to deal with multiview data to infer one view from another one and/or to improve classification systems. To do so nonlinear variants have been proposed such as Tang et al. (2017); Li et al. (2015). Our approach is based on the same assumptions as these methods i.e. each view is generated based on one common latent factor describing the "content" of the object and with its own "view" latent factor responsible for the difference between two observations. The main difference comes from the way the model is learned: while IBFA methods usually rely on regularization terms and on particular factorization functions to capture these factors (e.g. Damianou et al. (2012)), our model is much simpler (allowing scaling to large datasets) and makes use of a discriminator function to capture common and specific information based on a pair of observations.

Another family of related methods casts the problem as a domain transfer task. Different views are considered as different domains, and the problem becomes to project any image from a source domain to a target domain. Most of those approaches combine a prediction or auto-encoding loss ($\ell_1$) with an adversarial loss that is tasked to enforce a good visual quality, a technique used in Isola et al. (2016). The discriminators also serve to ensure that the produced output is in the correct domain, meaning that a discriminator must be learned for each domain. However, those methods are powerful as they can discover an alignment between two unpaired datasets, as shown in Kim

et al. (2017a); Zhu et al. (2017); Liu et al. (2017) using a cycle consistent auto-encoding loss. A third family of models, perhaps closer to our work, considers the problem of editing images based on manipulating the attributes. In this setting, most of the models consider a learning dataset where some factors of variation are labeled for each sample. Those labels can be used to disentangle between content and view. For example, the model proposed in Lample et al. (2017) trains an auto-encoder simultaneously with a discriminator used to remove the labeled information at a latent level. The model presented in Zhao et al. (2017) uses a variational auto-encoder framework instead. The attribute is given as a word in input and the disentanglement is ensured by a conditional discriminator at output level. The model proposed in Kim et al. (2017b) revisits the cycle approach by learning to generate outputs images with a given set of attribute values, and then to go back to the initial image.

In those last approaches, domain transfer and attribute manipulation, while GAN is used to ensure visual quality, most approaches are not generative in the sense that one input in the source domain always produces the same output in the target domain. Also, in these settings, one usually makes the assumption that the number of domains (or varying attributes) is very limited, as additional networks must be trained for each new domain. Our approach is original as we don't use information on the views, but instead, we just use the fact that two training samples represent the same content. This allows our approach to handle continuous view and content latent spaces, and thus to generate as many contents as needed and any number of views over these contents.

Other works have aimed at disentangling content and view/style without any supervision, i.e. based on unlabeled samples. In the Info-GAN model (Chen et al. (2016)), a code is passed along a random noise to generate an image. An encoder is then used to retrieve the code so that the mutual information between the generated image and the code is maximized. The beta-VAE model proposed in Higgins et al. (2016) is based on the VAE model, where the dimensions of the latent representation are forced to be as independent as possible using a strong regularization. The independent dimensions can then correspond to independent factors of the outputs. For these two models, the ability to disentangle content and view is not obvious since there is no supervision that can guide the learning. More specifically, these models disentangle low-level visual attributes but struggle to grasp higher level concepts.

The work closest to ours is the model proposed in Mathieu et al. (2016) in which they use an encoder to extract both view and content vectors from a datapoint and a decoder that combines both vectors to produce an output. They use both a reconstruction loss and a discriminator that serves multiple purpose. However, this work is mostly centered on disentangling the factors, and their purely generative abilities are limited.

## 7  CONCLUSION

We have proposed a generative model operating on a disentangled latent space which may be learned from multiview data without any view supervision, allowing its application to many multiview dataset. Our model allows generating realistic data with a rich view diversity. We also proposed a conditional version of this model which allows generating new views of an input image which may again be learned without view supervision. Our experimental results show the quality of the produced outputs, and the ability of the model to capture content and view factors. They also illustrate the ability of GMV and CGMV to capture the diversity over the CelebA dataset, and to generate more realistic samples than baseline models. In a near future, we plan to investigate the use of such an approach for data augmentation. Indeed, when only a few training data are available, one elegant solution for learning a model could be to generate new views of the existing data in order to increase the size of the training set. This solution will be explored in both semi-supervised and one-shot/few-shot learning settings.

ACKNOWLEDGMENTS

This work was supported by the French project LIVES ANR-15-CE23-0026-03.

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

APPENDIX

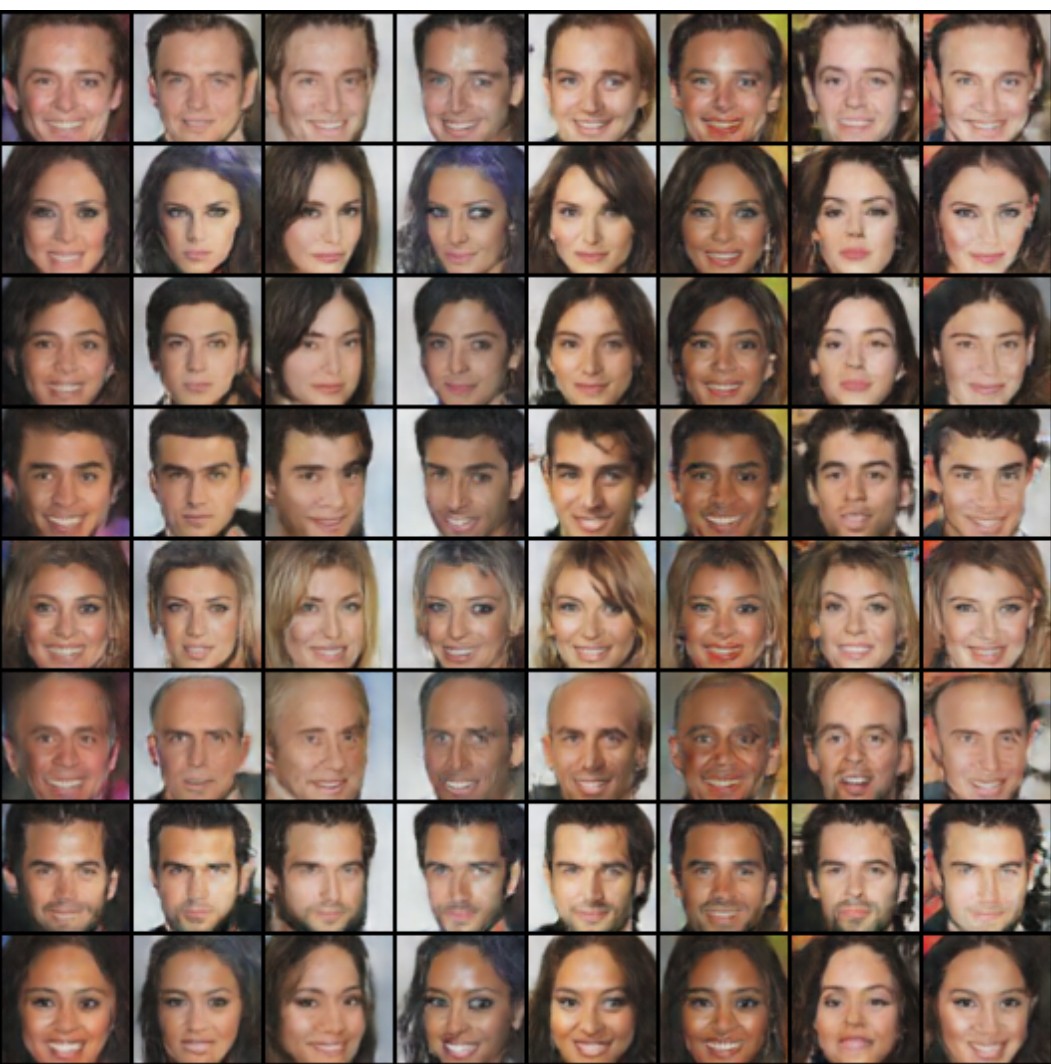

Figure 9: Additional results on GMV: All images in a row have been generated with the same content vector, and all images in a column have been generated with the same view vector

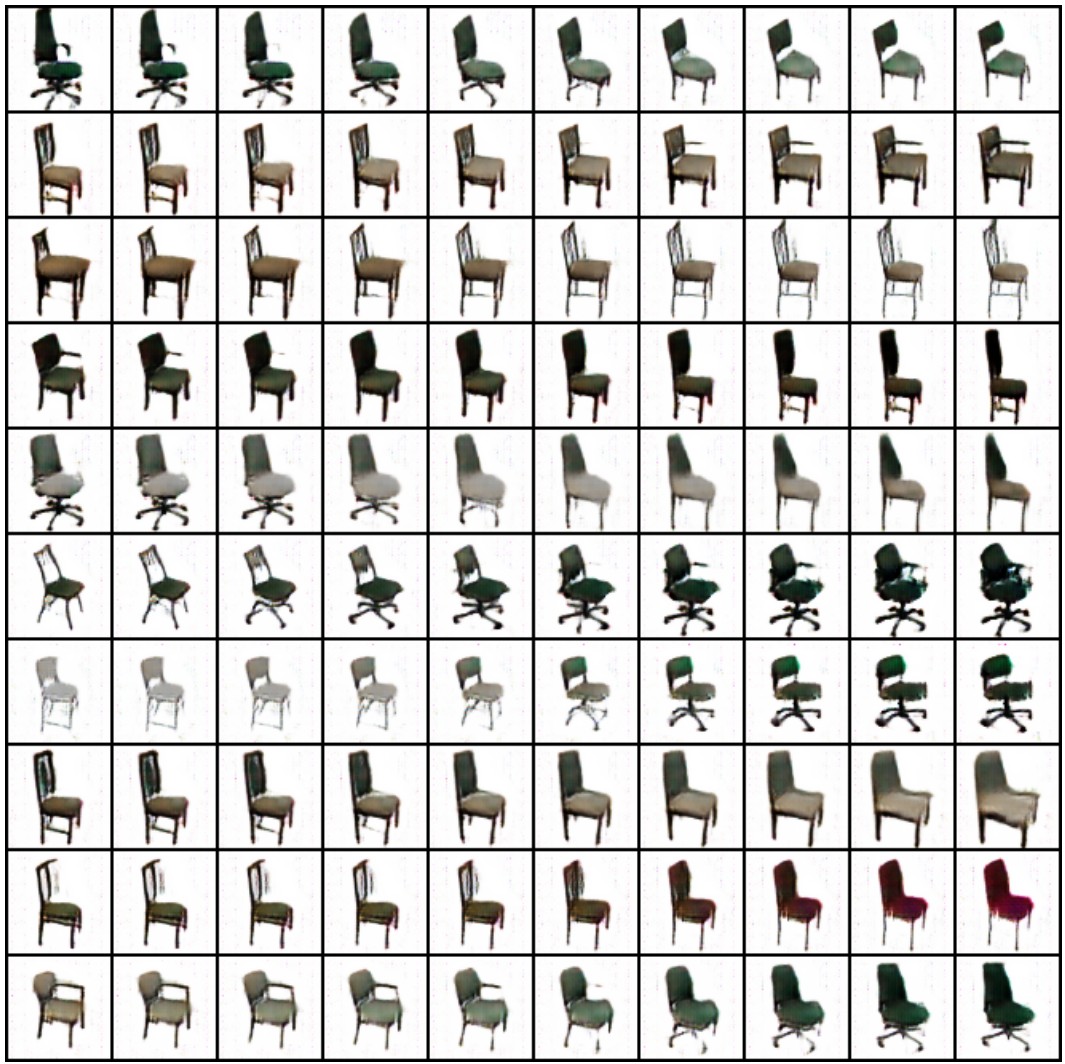

Figure 10: Interpolation GMV: The view is shared among each image. Content is interpolated.

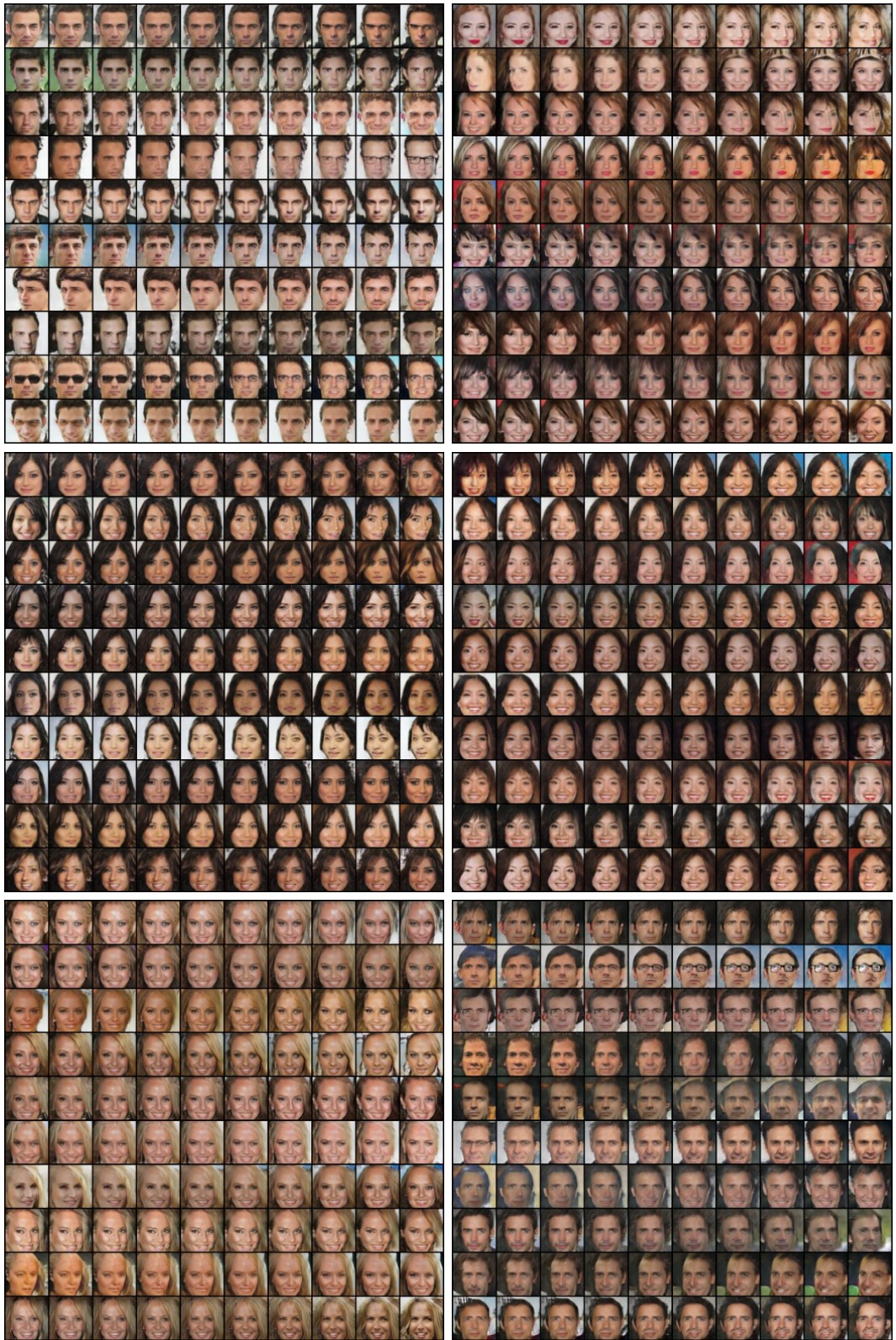

Figure 11: Additional interpolation on GMV: Each block is generated with the same content vector. The left columns and right columns are generated views. Images in between are interpolated

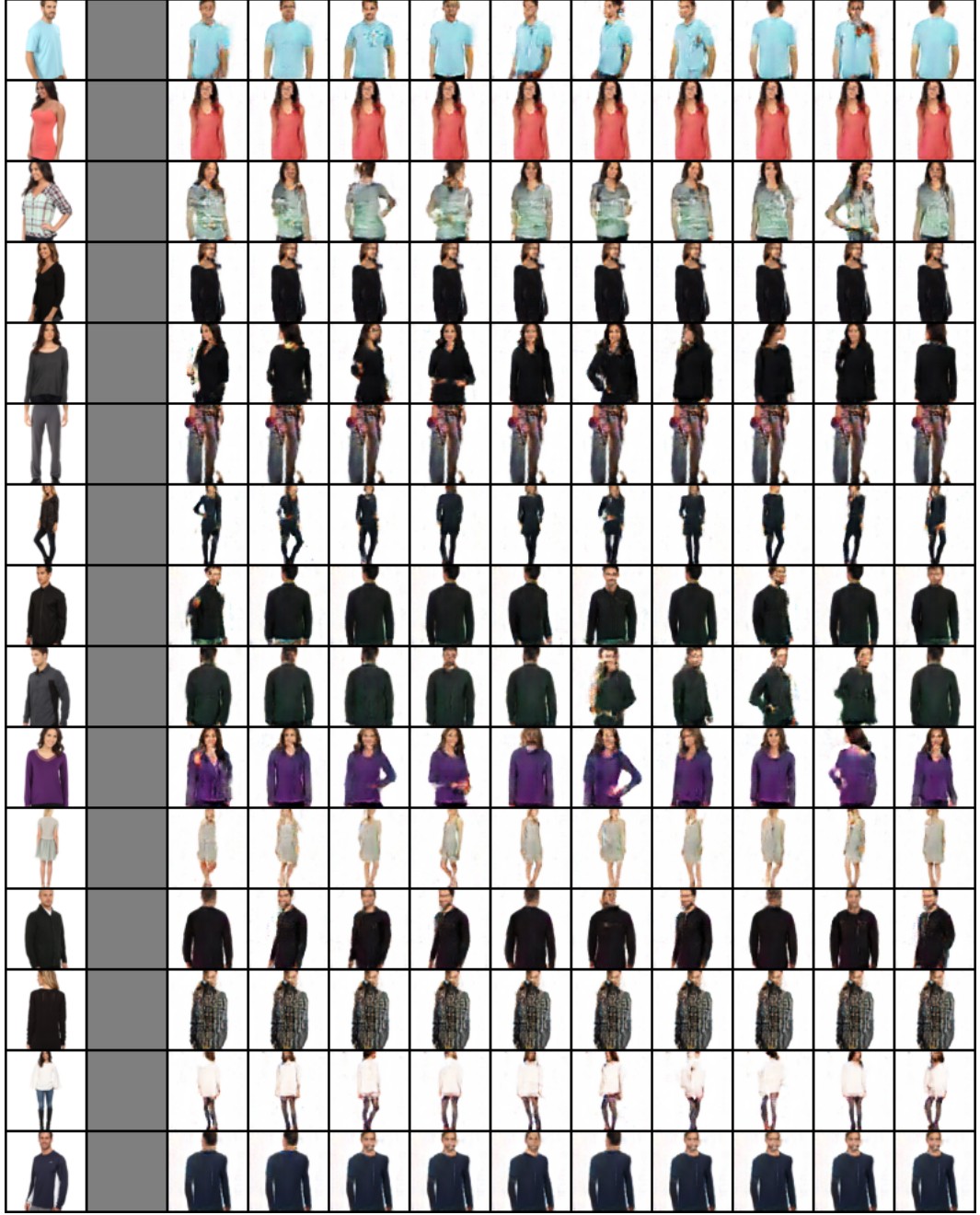

Figure 12: Additional results on C-GMV on the MVC cloth dataset

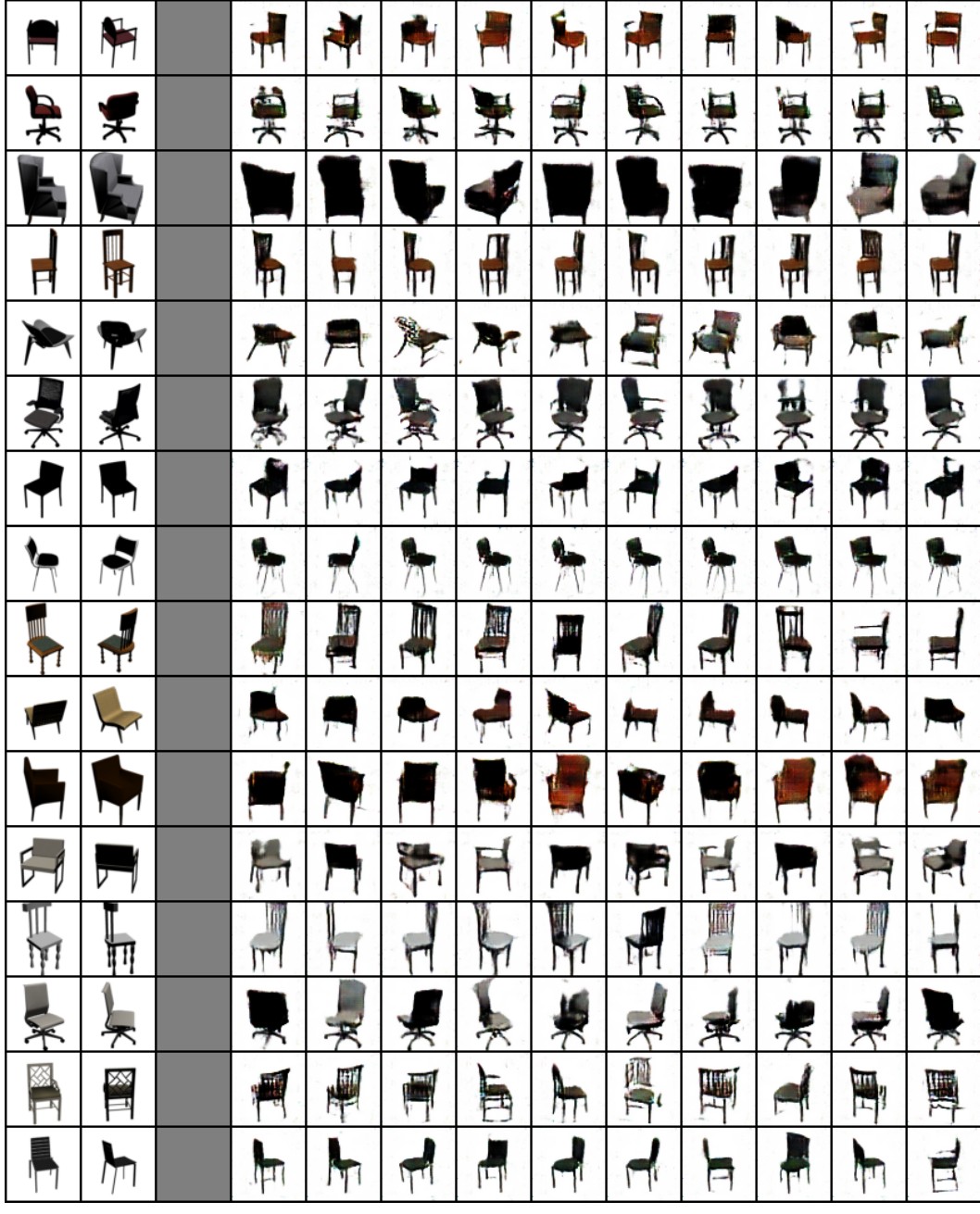

Figure 13: Additional results on C-GMV on the 3d chairs dataset

| Attribute | CGAN | GANx2 | GANx4 | GANx8 | Mathieu et al. | Mathieu et al. (DC-GAN) | GMV | CGMV | CelebA | Real Dist. |
|---|---|---|---|---|---|---|---|---|---|---|
| 5o Clock Shadow | 0.0820 | 0.0041 | 0.0098 | 0.2302 | 0.0000 | 0.0003 | 0.0077 | 0.0048 | 0.0103 | 0.1117 |
| Arched Eyebrows | 0.0040 | 0.1196 | 0.0608 | 0.0000 | 0.0003 | 0.1540 | 0.2152 | 0.2159 | 0.2182 | 0.2666 |
| Attractive | 0.0008 | 0.2726 | 0.1484 | 0.0000 | 0.0016 | 0.2254 | 0.4721 | 0.4423 | 0.5961 | 0.5112 |
| Bags Under Eyes | 0.0050 | 0.0051 | 0.0188 | 0.0139 | 0.0000 | 0.0011 | 0.0191 | 0.0111 | 0.0374 | 0.2054 |
| Bald | 0.0000 | 0.0065 | 0.0029 | 0.0000 | 0.0005 | 0.0069 | 0.0074 | 0.0021 | 0.0053 | 0.0227 |
| Bangs | 0.1542 | 0.0767 | 0.0728 | 0.0755 | 0.0000 | 0.0177 | 0.1128 | 0.1534 | 0.1903 | 0.1505 |
| Big Lips | 0.0005 | 0.0104 | 0.0069 | 0.0000 | 0.0000 | 0.0011 | 0.0206 | 0.0238 | 0.0347 | 0.2406 |
| Big Nose | 0.0209 | 0.0119 | 0.0316 | 0.0405 | 0.0000 | 0.0013 | 0.0254 | 0.0138 | 0.0437 | 0.2363 |
| Black Hair | 0.4733 | 0.1131 | 0.1704 | 0.1615 | 0.0246 | 0.1302 | 0.2297 | 0.2296 | 0.3937 | 0.2372 |
| Blond Hair | 0.0119 | 0.1201 | 0.0761 | 0.0000 | 0.0071 | 0.0698 | 0.1456 | 0.1074 | 0.1008 | 0.1487 |
| Blurry | 0.9984 | 0.5249 | 0.8238 | 0.9857 | 0.9995 | 0.7405 | 0.2895 | 0.3722 | 0.0861 | 0.0510 |
| Brown Hair | 0.0013 | 0.0014 | 0.0104 | 0.0000 | 0.0000 | 0.0000 | 0.0272 | 0.0225 | 0.0589 | 0.2061 |
| Bushy Eyebrows | 0.0135 | 0.0156 | 0.0296 | 0.0000 | 0.0000 | 0.0042 | 0.0338 | 0.0228 | 0.0647 | 0.1420 |
| Chubby | 0.0003 | 0.0011 | 0.0024 | 0.0000 | 0.0000 | 0.0013 | 0.0040 | 0.0011 | 0.0118 | 0.0576 |
| Double Chin | 0.0000 | 0.0004 | 0.0010 | 0.0000 | 0.0000 | 0.0000 | 0.0020 | 0.0011 | 0.0029 | 0.0471 |
| Eyeglasses | 0.0029 | 0.0393 | 0.0502 | 0.0004 | 0.0000 | 0.0034 | 0.0381 | 0.0257 | 0.0384 | 0.0655 |
| Goatee | 0.0045 | 0.0070 | 0.0244 | 0.0038 | 0.0000 | 0.0003 | 0.0225 | 0.0061 | 0.0282 | 0.0633 |
| Gray Hair | 0.0000 | 0.0023 | 0.0070 | 0.0005 | 0.0000 | 0.0016 | 0.0089 | 0.0011 | 0.0021 | 0.0426 |
| Heavy Makeup | 0.0013 | 0.1304 | 0.0976 | 0.0000 | 0.0003 | 0.1317 | 0.3336 | 0.3085 | 0.3897 | 0.3869 |
| High Cheekbones | 0.0243 | 0.2491 | 0.2169 | 0.0000 | 0.1098 | 0.3209 | 0.3494 | 0.3892 | 0.3374 | 0.4568 |
| Male | 0.4862 | 0.3711 | 0.4102 | 0.8821 | 0.4032 | 0.2135 | 0.3054 | 0.2354 | 0.3071 | 0.4185 |
| Mouth Slightly Open | 0.0614 | 0.3391 | 0.2964 | 0.2042 | 0.0841 | 0.2206 | 0.3926 | 0.3918 | 0.3621 | 0.4846 |
| Mustache | 0.0013 | 0.0056 | 0.0126 | 0.0000 | 0.0000 | 0.0000 | 0.0182 | 0.0053 | 0.0163 | 0.0417 |
| Narrow Eyes | 0.0159 | 0.0214 | 0.0276 | 0.0000 | 0.0212 | 0.0595 | 0.0786 | 0.0944 | 0.0816 | 0.1126 |
| No Beard | 0.6198 | 0.9513 | 0.8631 | 0.1651 | 0.9952 | 0.9976 | 0.9360 | 0.9669 | 0.9408 | 0.8341 |
| Oval Face | 0.0000 | 0.0079 | 0.0008 | 0.0000 | 0.0000 | 0.0042 | 0.0416 | 0.0296 | 0.1074 | 0.2842 |
| Pale Skin | 0.0040 | 0.0111 | 0.0115 | 0.0000 | 0.0000 | 0.0061 | 0.0195 | 0.0352 | 0.0303 | 0.0426 |
| Pointy Nose | 0.0000 | 0.0166 | 0.0520 | 0.0000 | 0.0013 | 0.0280 | 0.1700 | 0.1085 | 0.1363 | 0.2788 |
| Receding Hairline | 0.0000 | 0.0472 | 0.0290 | 0.0000 | 0.1270 | 0.0839 | 0.0521 | 0.0217 | 0.0292 | 0.0802 |
| Rosy Cheeks | 0.0000 | 0.0000 | 0.0001 | 0.0000 | 0.0000 | 0.0000 | 0.0034 | 0.0011 | 0.0050 | 0.0661 |
| Sideburns | 0.0442 | 0.0024 | 0.0178 | 0.1187 | 0.0000 | 0.0000 | 0.0123 | 0.0032 | 0.0155 | 0.0569 |
| Smiling | 0.1418 | 0.4043 | 0.3956 | 0.0829 | 0.3151 | 0.4058 | 0.4606 | 0.4979 | 0.3953 | 0.4830 |
| Straight Hair | 0.0003 | 0.0864 | 0.0138 | 0.0000 | 0.0132 | 0.0804 | 0.0984 | 0.0672 | 0.2882 | 0.2073 |
| Wavy Hair | 0.2595 | 0.0044 | 0.1663 | 0.0984 | 0.0000 | 0.0000 | 0.1111 | 0.1082 | 0.0939 | 0.3200 |
| Wearing Earrings | 0.0000 | 0.0013 | 0.0001 | 0.0000 | 0.0000 | 0.0101 | 0.0209 | 0.0071 | 0.0321 | 0.1893 |
| Wearing Hat | 0.0021 | 0.0261 | 0.0312 | 0.0000 | 0.0000 | 0.0029 | 0.0324 | 0.0283 | 0.0426 | 0.0484 |
| Wearing Lipstick | 0.0127 | 0.2923 | 0.2287 | 0.0000 | 0.0132 | 0.3156 | 0.4738 | 0.4971 | 0.5568 | 0.4715 |
| Wearing Necklace | 0.0000 | 0.0000 | 0.0000 | 0.0000 | 0.0000 | 0.0000 | 0.0011 | 0.0008 | 0.0003 | 0.1231 |
| Wearing Necktie | 0.0000 | 0.0080 | 0.0024 | 0.0000 | 0.0000 | 0.0053 | 0.0095 | 0.0026 | 0.0153 | 0.0733 |
| Young | 0.9183 | 0.9068 | 0.8170 | 0.4186 | 0.9307 | 0.9214 | 0.8941 | 0.9328 | 0.9292 | 0.7723 |

Table 8: Distribution of the different attributes over generated samples. For example, 3.8 % of the samples generated by the GMV model exhibit the "Eyeglasses" attribute.

