# OpenReview forum: "Multi-View Data Generation Without View Supervision"
_ICLR.cc/2018/Conference — Accept (Poster)_

### Official Review · AnonReviewer1 · 2017-11-23
**The authors propose a GAN formulation for multi-view learning trained to disentangle content from other aspects ("view") influencing the image, presented from a slightly narrow perspective.**

**Rating:** 7
**Confidence:** 3

**Review:**

The paper proposes a GAN-based method for image generation that attempts to separate latent variables describing fixed "content" of objects from latent variables describing properties of "view" (all dynamic properties such as lighting, viewpoint, accessories, etc). The model is further extended for conditional generation and demonstrated on a range of image benchmark data sets.

The core idea is to train the model on pairs of images corresponding to the same content but varying in views, using adversarial training to discriminate such examples from generated pairs. This is a reasonable procedure and it seems to work well, but also conceptually quite straightforward -- this is quite likely how most people working in the field would solve this problem, standard GAN techniques are used for training the generator and discriminator, and the network architecture is directly borrowed from Radford et al. (2015) and not even explained at all in the paper. The conditional variant is less obvious, requiring two kinds of negative images, and again the proposed approach seems technically sound.

Given the simplicity of the algorithmic choices, the potential novelty of the paper lies more in the problem formulation itself, which considers the question of separating two sets of latent variables from each other in setups where one of them (the "view") can vary from pair to pair in arbitrary manner and no attributes characterising the view are provided. This is an interesting problem setup, but not novel as such and unfortunately the paper does not do a very good job in putting it into the right context. The work is contrasted only against recent GAN-based image generation literature (where covariates for the views are often included) and the aspects related to multi-view learning are described only at the level of general intuition, instead of relating to the existing literature on the topic. The only relevant work cited from this angle is Mathieu et al. (2016), but even that is dismissed lightly by saying it is worse in generative tasks. How about the differences (theoretical and empirical) between the proposed approach and theirs in disentangling the latent variables? One would expect to see more discussion on this, given the importance of this property as motivation for the method.

The generative story using three sets of latent variables, one shared, to describe a pair of objects corresponds to inter-battery factor analysis (IBFA) and is hence very closely related to canonical correlation analysis as well (Tucker "An inter-battery method of factor analysis", Psychometrika, 1958; Klami et al. "Bayesian canonical correlation analysis", JMLR, 2013). Linear CCA naturally would not be sufficient for generative modeling and its non-linear variants (e.g. Wang et al. "Deep variational canonical correlation analysis", arXiv:1610.03454, 2016; Damianou et al. "Manifold relevance determination", ICML, 2012) would not produce visually pleasing generative samples either, but the relationship is so close that these models have even been used for analysing setups identical to yours (e.g. Li et al. "Cross-pose face recognition by canonical correlation analysis", arXiv:1507.08076, 2015) but with goals other than generation. Consequently, the reader would expect to learn something about the relationship between the proposed method and the earlier literature building on the same latent variable formulation. A particularly interesting question would be whether the proposed model actually is a direct GAN-based extension of IBFA, and if not then how does it differ. Use of adversarial training to encourage separation of latent variables is clearly a reasonable idea and quite likely does better job than the earlier solutions (typically based on some sort of group-sparsity assumption in shared-private factorisation) with the possible or even likely exception of Mathieu at al. (2016), and aspects like this should be explicitly discussed to extend the contribution from pure image generation to multi-view literature in general.

The empirical experiments are somewhat non-informative, relying heavily on visual comparisons and only satisfying the minimum requirement of demonstrating that the method does its job. The results look aesthetically more pleasing than the baselines, but the reader does not learn much about how the method actually behaves in practice; when does it break down, how sensitive it is to various choices (network structure, learning algorithm, amount of data,  how well the content and view can be disentangled from each other, etc.). In other words, the evaluation is a bit lazy somewhat in the same sense as the writing and treatment of related work; the authors implemented the model and ran it on a collection of public data sets, but did not venture further into scientific reporting of the merits and limitations of the approach.

Finally, Table 1 seems to have some min/max values the wrong way around.


Revision of the review in light of the author response:
The authors have adequately addressed my main remarks, and while doing so have improved both the positioning of the paper amongst relevant literature and the somewhat limited empirical comparisons. In particular, the authors now discuss alternative multi-view generative models not based on GANs and the revised paper includes considerably extended set of numerical comparisons that better illustrate the advantage over earlier techniques. I have increased my preliminary rating to account for these improvements.

---

> ### Author Response · Authors · 2017-12-31
> **Re: The authors propose a GAN formulation for multi-view learning trained to disentangle content from other aspects ("view") influencing the image, presented from a slightly narrow perspective.**
>
> We thank the reviewer for the comments and feedback. We apologize for the late reply due to the large number of experiments that have been made to improve the quality of the paper.
>
> Concerning the fact that our generative model is “conceptually quite straightforward”, we would like to emphasis that the proposed paper is as far as we know the first paper to evaluate this idea of using a discriminator on pairs of outputs for the multiview problem, this discriminator being in charge of telling is the two outputs correspond to the same object.
>
> We acknowledge the reviewer for pointing us this extensive literature on IBFA and on similar ideas in CCA and non linear variants of CCA. Of course our method is clearly related to this literature and we added this related work on the state if the art section. As suggested by the reviewer the assumption made by our method is very similar to the one made with IBFA models. The main difference being in the way the models are learned: by using ‘strong‘ regularization and particular factorization functions in the IBFA literature, or by using a discriminator in our case.  Note also that most experiments in the IBFA literature are based on datasets where a limited finite number of possible views is provided while our model is evaluated on complex datasets with multiple possible views, without any available view supervision. A detailed discussion on this point has been added in Section 6.
>
> About Radford architecture. Yes we do reuse the architecture in [Radford et al., 2015] for the DCGAN architecture because the core idea of the paper is elsewhere, as it was the case for [Mathieu et al., 2016]. Actually the main features of our method, its ability to learn from data whose views are not aligned between objects and which are unlabeled comes from our particular learning scheme and the way we build pairs of examples. This is why we focus the presentation of our method on this particular way of constructing training examples for our models.
>
> Please consider  that we have added a large additional experimental section that objectively evaluates the quality of the generated samples of the different models (GMV, CMGV, GANx, CGAN and Mathieu et al.) in terms of quality of the outputs, and in terms of diversity of the generated samples, showing the superiority of our model w.r.t these baselines (new section 5.3, pages 11 to 13 of the new version)

---

### Official Review · AnonReviewer2 · 2017-11-26
**Interesting approach on separating content and views for a specific distribution, but lacks interpretability for diverse datasets**

**Rating:** 5
**Confidence:** 4

**Review:**

This paper firstly proposes a GAN architecture that aim at decomposing the underlying distribution of a particular class into "content" and "view". The content can be seen as an intrinsic instantiation of the class that is independent of certain types of variation (eg viewpoint), and a view is the observation of the object under a particular variation. The authors additionally propose a second conditional GAN that learns to generate different views given a specific content.

I find the idea of separating content and view interesting and I like the GMV and CGMV architectures. Not relying on manual attribute/class annotation for the views is also positive. The approach seems to work well for a relatively clean setup such as the chair dataset, but for the other datasets the separation is not so apparent. For example, in figure 5, what does each column represent in terms of view? It seems that it depends heavily on the content. That raises the question of how useful it is to have such a separation between content and views; for some datasets their diversity can be a bottleneck for this partition, making the interpretation of views difficult.

A missing (supervised) reference that considers also the separation of content and views.
[A] Learning to generate chairs with convolutional neural networks, Alexey Dosovitskiy, Jost Tobias Springenberg, Thomas Brox, CVPR 15

Q:Figure 5, you mean "all images in a column were generated with the same view vector"
Q: Why on Figure 7 you use different examples for CGAN?

---

> ### Author Response · Authors · 2017-12-31
> **Re: Interesting approach on separating content and views for a specific distribution, but lacks interpretability for diverse datasets**
>
> We thank the reviewer for the comments and feedback. We apologize for the late reply due to the large number of experiments that have been made to improve the quality of the paper.
>
> As far as we understand, the main concern is about the fact that the interpretation of the notion of view can be difficult depending on the nature of the dataset. We agree on that point. Indeed, what we call ‘content’ in this paper corresponds to the invariant factors contained in a set of images representing a same object, the view corresponding to the remaining ‘changing’ factors. This is the assumption also made for the IBFA and CCA based approaches (see next review). We have added a discussion on this point in the paper in the literature review section. Note also  that the more difficult interpretation of views in our work is the counterpart of  the increased ability of the method to deal with various datasets.
>
> Concerning the suggested reference, our related work is focused on  models that are not based on view supervision.
>
> Note that we have added a large additional experimental section that objectively evaluates the quality of the generated samples of the different models (GMV, CMGV, GANx, CGAN and Mathieu et al.) in terms of quality of the outputs, and in terms of diversity of the generated samples, showing the superiority of our model w.r.t these baselines (new section 5.3, pages 11 to 13 of the new version)

---

### Official Review · AnonReviewer3 · 2017-11-30
**A good paper using GANs for multi-view image generation**

**Rating:** 7
**Confidence:** 5

**Review:**

The paper proposes a new generative model based on the Generative Adversarial Network (GAN). The method disentangles the content and the view of objects without view supervision. The proposed Generative Multi-View (GMV) model can be considered to be an extension of the traditional GAN, where the GMV takes the content latent  vector and the view latent vector as input. In addition, the GMV is trained to generate a pair of objects that share the content but with different views. In this way, the GMV successfully models the content and the view of the objects without using view labels. The paper also extends GMV into a conditional generative model that takes an input image and generates different views of the object in the input image. Experiments are conducted on four different datasets to show the generative ability of the proposed method.

Positives:
- The proposed method is novel in disentangling the content and the view of objects in a GAN and training the GAN with pairs of objects. By using pairs that share the content but with different views, the model can be trained successfully without using view labels.

- The experimental results on the four datasets show that the proposed network is able to model the context and the view of objects when generating images of these objects.

Negatives:
- The paper only shows comparison between the proposed method and several baselines: DCGAN and CGAN. There is no comparison with methods that also disentangle the content from the view such as Mathieu et al. 2016.

- For the comparison with CGAN in Figure 7, it would be better to show the results of C-GMV and CGAN on the same input images. Then it is easier for the readers to see the differences in the results from the two methods.

---

> ### Author Response · Authors · 2017-12-31
> **Re: A good paper using GANs for multi-view image generation**
>
> We thank the reviewer for the comments and feedback. We apologize for the late reply due to the number of additional experiments that have been made to improve the quality of the paper.
>
> The first concern of the reviewer is about the lack of comparisons with other techniques. We updated the paper with results obtained on the same tasks with the approach by Mathieu et al. 2016 which is the closest to ours. Note that we were able to obtain comparable quality of outputs using the Mathieu et al. model by carefully testing many different neural networks architectures, the ones being provided in the open-source implementation, provided by the authors being inefficient on our problems. The quality of the generated samples of the different models (GMV, CMGV, GANx, CGAN and Mathieu et al.) have been evaluated in terms of quality of the outputs, and in terms of diversity of the generated samples, showing the superiority of our model w.r.t these baselines (new section 5.3, pages 11 to 13 of the new version)
>
> We have also taken care to illustrate samples of the different models based on the same input images to allow for a better qualitative comparison  (Figure 8)

---

### Decision · Program_Chairs · 2018-01-29
**ICLR 2018 Conference Acceptance Decision**

**Decision:**

Accept (Poster)

**Comment:**

This paper presents an unsupervised GAN-based model for disentagling the multiple views of the data and their content.

Overall it seems that this paper was well received by the reviewers, who find it novel and significant . The consensus is that the results are promising.

There are some concerns, but the major ones listed below have been addressed in the rebuttal. Specifically:
-	R3 had a concern about the experimental evaluation, which has been addressed in the rebuttal.
-	R2 had a concern about a problem inherent in this setting (what is treated as “content”), and the authors have clarified in the discussion the assumptions under which such methods operate.
-	R1 had concerns related to how the proposed model fits in the literature. Again, the authors have addressed this concern adequately.